# The Hybridization Barrier between Herbaceous *Medicago sativa* and Woody *M. arborea* Is Weakened by Reproductive Abnormalities in *M. sativa* Seed Parents

**DOI:** 10.3390/plants12040962

**Published:** 2023-02-20

**Authors:** Edwin Bingham, John Irwin

**Affiliations:** 1Agronomy Department, University Wisconsin, Madison, WI 53706, USA; 2School of Agriculture and Food Sciences, University Queensland, Brisbane, QLD 4072, Australia

**Keywords:** species, interspecific hybrids, gametes, unreduced gametes, hybrid breakdown, plant breeding

## Abstract

Historically, crosses between *Medicago sativa* (alfalfa) and *M. arborea* with alfalfa as the seed parent failed, as did crosses using *M. arborea* as the seed parent. Thus, a reproductive barrier kept the two species isolated until early in this century. The breakthrough came when alfalfa seed parents were identified in Wisconsin USA and Queensland AU that produced partial hybrids (hereafter hybrids). The hybrids were obtained by making large numbers of crosses on selected alfalfa parents. This was the first level of weakening the crossing barrier as reported in *Plants* in 2013. Further weakening of the barrier is reported herein whereby more hybrids were obtained with fewer crosses. This was accomplished by pedigree selection for new alfalfa seed parents and by using a product of the first hybrids called Alborea. New alfalfa seed parents were crossed with *M. arborea*, and Alborea parents were backcrossed to *M. arborea*. Hybrid plants were produced with fewer crosses in both cases. These hybrids, like the first hybrids, have mostly alfalfa traits but also have traits from *M. arborea*. It was theorized early on that the alfalfa component could be explained by 2n eggs in the alfalfa parents that were fertilized by normal n gametes from *M. arborea*. Evidence that the Wisconsin alfalfa and Alborea seed parents did in fact produce 2n eggs was reported in *Plants* in 2022. Moreover, they produced 2n eggs at approximately the same frequency that they produced hybrids. As reported herein, Alborea parents produced the highest frequency of hybrids and thus had the weakest barrier. Importantly, they also have the highest frequency of 2n eggs. It was determined that alfalfa and Alborea parents that produce 2n eggs and hybrids, also produce 2n pollen. In effect, an experiment was undertaken in reverse showing that 2n pollen could be used to screen for plants that produce hybrids. In the thousands of crosses made over the years, fertilization of normal n eggs in alfalfa parents always failed. Normal meiosis appears to be the main barrier to producing interspecific hybrids in our case. Fertilization of abnormal 2n eggs ensures sufficient alfalfa genetic material to continue embryogenesis. Evidently, the meiotic abnormality of 2n eggs is the major factor that weakens the crossing barrier.

## 1. Introduction

This paper is a follow-up of a 2013 report in *Plants* [1] that summarized the first hybrids of *Medicago sativa* X *M. arborea* by sexual crosses. *Medicago arborea* is a large woody perennial with many traits for potential use in the improvement of alfalfa (*M. sativa*). This was the incentive to search for alfalfa parents that would cross with *M. arborea* and produce the hybrids required for trait transmission. Yield is a key trait of interest because alfalfa suffers yield stagnation [2,3,4] and could benefit from introgression of exotic germplasms. The first level of weakening the crossing barrier came when alfalfa seed parents were found in Wisconsin and Queensland that produced a few hybrids after many crosses [1]. The two programs produced more than 30 hybrids by 2013. The hybrids have been used to transfer anthracnose resistance to alfalfa [5], shown to have heterosis for herbage yield in crosses with alfalfa [6,7], and are being used to restructure the morphology of alfalfa [8]. In addition, hybrids have been intercrossed to produce a cultigen named Alborea [8]. Alborea crosses with itself and with alfalfa and is a bridge for the transfer of *M. arborea* traits.

When the first hybrids were obtained, it was obvious that reproductive factors were involved, and several reproductive factors were examined [1]. They included determining if a normal seed from occasional self-pollination aided retention of pods with a hybrid seed. This was not the case as most hybrids came from single seeded pods. Pollen mentoring and other pollination factors were also ruled out, leaving the focus on female gametogenesis.

In 2013 it was not known that plants with an even weaker barrier would be found in the next few years. Further weakening of the hybridization barrier reported here is based on finding new seed parents that produce more hybrids with fewer crosses. The new hybrids, like the first hybrids, have mostly alfalfa traits and DNA but also have *M. arborea* traits and DNA [5]. It was hypothesized early in the work that fertilization of 2n eggs could explain the alfalfa base in the first hybrids [5]. The partial hybrid condition could then be explained by elimination of some *M. arborea* genetic material during embryogenesis. A follow-up study reported in *Plants* in 2022 [9] found that the alfalfa seed parents that produced hybrids did in fact produce 2n eggs along with mostly normal n eggs. Importantly, 2n eggs were produced at the same frequency as hybrids. A test for 2n eggs reported herein involves new Wisconsin seed parents with further weakening of the crossing barrier.

There are examples in other species of partial hybrids where 2n female gametes have been suggested to explain the predominance of the female genome in hybrids. This happened in Tripsacum austral X Zea mays and the hybrids were called “counterfeit hybrids” [10]. In a cross of *Brassica napus X Orychophragmus, violaceus* hybrids contained the female genome and 2n female gametes were suggested to explain this [11]. Partial hybrids involve chromosome elimination, which has been studied extensively in *Hordeum*, where asynchronous mitotic cell cycles are involved [12].

The connection between weakening the crossing barrier and reproductive abnormalities began with seed parent M8 reported in 2013 [1]. Seed parent M8 was derived from subspecies crosses and had a weaker barrier than the pure alfalfa parents. Moreover, M8 also had conspicuous reproductive abnormalities in pollen and seed production. Clone M8 was a subspecies hybrid involving *M*. *sativa* and its subspecies *coerulea* and *falcata* in a complex three-way X three-way cross [1]. Subspecies parents were all normal in male and female fertility, whereas M8 has only a trace of pollen and half of the normal seed set. In addition to these visual abnormalities, M8 produces 2n eggs as reported in 2022 [9]. It was apparent that M8 suffered hybrid breakdown of reproductive factors due to the diversity of the parents. In fact, the degree of the hybrid breakdown indicates the level of diversity of the subspecies rivals that of a species. Since M8 had a first level of hybrid breakdown when used to produce hybrids, those hybrids could be expected to have a second level and further weakening of the crossing barrier. This was the case, and in the next few years several plants with further weakening were found and are the subject of this paper.

## 2. Results and Discussion

### 2.1. Further Weakening of the Crossing Barrier

Hybrid frequencies are reported in Table 1 for comparison of the weakened crossing barrier reported from 2003–2013 [1] and for the further weakened barrier evidenced by the smaller number of pollinations required to produce a hybrid from 2013–2022. Alfalfa parents that did not produce hybrids in Wisconsin and Queensland serve as checks and are also reported in Table 1. They received almost twice as many pollinations per plant as the hybrid producers yet produced no hybrids.

In the 2013 report, the alfalfa parents with a weakened barrier required pollinating 420 florets to obtain a hybrid in Queensland and 250 florets in Wisconsin. Parent M8 (see Section 3.2) in Wisconsin was from a complex cross of *M. sativa* subspecies and required only 85 florets to produce a hybrid. In addition to being the most efficient hybrid producer at that time, M8 was the only parent in the 2013 study that had observable reproductive abnormalities [1]. Reproductive abnormalities in pollen and seed production, and the production 2n eggs are a recurring theme in the following results.

The period with further weakening from 2013–2022 in Queensland included four alfalfa plants as checks that produced no hybrids, and an S1 progeny of M8, WA2570, that produced hybrids (Table 1). In fact, the M8 S1–WA2570, a pedigree selection, was a more efficient hybridizer than M8 and required only 38 florets pollinated to produce a hybrid. This is a good example of further weakening. Given that M8 had reproductive abnormalities including 2n eggs, and that these could be compounded by inbreeding depression in the S1, further weakening of M8 S1 could be due to more abnormalities. Evidence is mounting in the results that further weakening is due to reproductive abnormalities.

In the 2013–2022 period in Wisconsin (Table 1) alfalfa parents 6–4 ms and M8 used in the 2003–2013 period, were used again as checks with roughly the same results in both periods. From 2013–2022 in Wisconsin, selected Alborea seed parents (see Section 3.4) were crossed with *M. arborea* pollen to test their hybrid production. The Alborea 101 plant required only 13 florets pollinated to produce a hybrid, while Alborea 301 plants 301a required 25, and 301b required 50. The *M. arborea* germplasm added to the genetic base of M8 likely increased hybrid breakdown of reproduction. The increase in 2n eggs is reported next in Section 2.2.

Turning to Alborea parents in the 2013–2022 period in Queensland (Table 1) WA Alborea 2554 showed no weakening and required 1000 florets to be pollinated to produce a hybrid. In contrast, WA Alborea 2973 X Sequel seed parent showed slight further weakening requiring 100 pollinations per hybrid.

We can only speculate about the difference in the Alborea parents in Queensland versus those in Wisconsin, as 2n egg tests could not be undertaken on Queensland parents. However, the Alborea seed parents used in Queensland were derived from alfalfa seed parent MBms, a normal alfalfa plant (see Section 3.1). MBms and its derivatives required many crosses with *M. arborea* to produce a hybrid. In contrast, the Wisconsin Alborea parents were derived from the clone M8 (see Section 3.2) that had abnormalities, including 2n eggs. Some M8 abnormalities would transfer to Alborea which would have additional hybrid breakdown in fertility. Thus, Alborea has low female fertility from sterile to half that of M8, and a higher frequency of 2n eggs. Among the abnormalities, the 2n eggs are the only categorical meiotic abnormality; however, impaired female fertility is evidence of unidentified meiotic abnormalities. Hence, the difference between Queensland and Wisconsin Alborea seed parents in hybrid production could be explained by more abnormalities in Wisconsin Alborea. The argument that the crossing barrier is weakened by reproductive abnormalities continues.

### 2.2. Evidence That 2n Eggs Enable Hybrid Production

Alfalfa and Alborea seed parents that were tested for 2n egg production are reported in Table 2. The test involved pollinating the 4x seed parents with pollen from 8x alfalfa as was undertaken in a previous study [9] and described in Section 3.5. The 4x X 8x cross produces mainly 6x embryos, most of which abort. Thus, little or no seed is produced on male sterile plants if they produce normal n eggs. If plants also produce 2n eggs, the occurrence of 2n eggs is confirmed by 8x progeny.

Results for alfalfa parents in Table 2 are from an earlier study [1] and are reported for comparison with Alborea parents in this study. In general, the frequency of 2n eggs and 8x progeny is low in the alfalfa plants. Alfalfa parent MBms had a lower frequency of 2n eggs than parent M8, at 5 versus 9 per 100 pollinations, and MBms is a less efficient hybridizer than M8. This association between the frequency of 2n eggs and hybrid production is also seen in the Alborea parents reported in Table 2.

Alborea seed parents (see Section 3.4) all produced a small amount of pollen; approximately 25% of normal alfalfa due to meiotic abnormalities. As expected, the pollen resulted in some seed from self-pollination, identified as 4x progeny by pollen size (Table 2). Abundant pods formed after only 50 florets were pollinated with 8x pollen, and crossing was discontinued. At the time it was assumed that there was too much self-seed produced to undertake the test. However, the grow-out revealed 8x progeny among the 4x self-progeny. Moreover, all Alborea parents produced 2n eggs as determined by the 8x progeny. Importantly, the 8x progeny were produced with only 50 florets pollinated. Alborea 101 plant and Alborea plants 301a and 301b produced hybrids and had 2n eggs, whereas plants 301c and 301d also had 2n eggs but were not tested for hybrid production. The latter were tested for 2n eggs to increase information about Alborea, and plant 301c had the highest frequency of 2n eggs in the test. Photomicrographs of 2n eggs in cleared ovules of alfalfa can be seen in a previous study [13]. The study found that 2n eggs in alfalfa were produced by restitution of the second meiotic division.

### 2.3. Evidence That 2n Pollen Can Be Used to Screen for Plants That Produce 2n Eggs and Hybrids: An Experiment Performed in Reverse Order

Testing for 2n eggs is laborious, requiring 8x alfalfa for pollination and growing progeny to maturity in order to test for the large pollen that identifies 8x plants. Testing for 2n pollen (see Section 3.7, Figure 1) is easy by comparison since it only requires screening the pollen of seed parents for normal n pollen and large 2n pollen. Thus, pollen of the seed parents was examined and large 2n pollen was observed (see Section 3.7, Figure 1, and Table 2). Large 2n pollen is a categorical trait and counts were not necessary. In effect, an experiment had been undertaken in reverse order and indicated that selection of seed parents for wide hybridization in this *Medicago* material could be achieved by screening for 2n pollen.

The 2n pollen results do not mean that the 2n egg research could have been bypassed in our case, because the presence of 2n eggs was required to explain the predominant alfalfa content of the hybrids [9]. It was fortunate that examination of pollen was the obvious next step while completing the 2n egg research. Henceforth, the presence of 2n pollen can be used to select seed parents for interspecific crosses involving the *Medicago* species in this study.

## 3. Materials and Methods

### 3.1. Alfalfa Seed Parents 2n = 4x = 32

Male sterile (ms) alfalfa seed parents were used in both Wisconsin and Queensland to minimize self-pollination in crosses with *M. arborea* and 8x alfalfa. Cross pollinations were undertaken in a greenhouse in all experiments. The alfalfa plants were of cultivated origin, and were described in earlier publications [1,5,6]. Plants were maintained as clones and were periodically propagated by shoot cuttings. Alfalfa clone MBms was the first alfalfa plant to produce hybrids in crosses of alfalfa X *M. arborea*. It was found among the progeny of cultivars Magnum III X Blaser XL. It is vigorous and has normal female fertility. Its male sterility is due to a cytoplasmic factor in mitochondrial DNA [14].

### 3.2. Special Clone M8 2n = 4x = 32

Clone M8 was described earlier [1,9], but a brief description is presented in Table 3 because M8 was the first seed parent to expose the link between meiotic abnormalities and hybrid production. This inspired the research on further weakening.

This crossing exercise was undertaken 20 years ago to mix the genomes of the crossable *M*. *sativa* subspecies and to study the effect on morphology and reproduction. The seed from the final cross is a cultigen named ‘Mixomatic’ [17]. Clone M8 was selected from a population of Mixomatic plants for use because it was self-sterile and had cream-colored flowers. The self-sterility minimizes self-pollination and maximizes the number of ovules available for pollination by *M. arborea*. The cream flower color enables identifying hybrids with *M. arborea* that have yellow flowers.

### 3.3. Medicago arborea Pollen Parents 2n = 4x = 32

*Medicago arborea* materials used in Wisconsin were from the Greek Islands and around the Mediterranean Sea (PI 199254, PI 564540 and PI 330677). Those used in Queensland were collected in Spain (Australian Pastures Genebank (APG) 30528). *M. arborea* is a woody perennial bush with yellow to orange flowers, flat coiled pods, and no crown, as described by Small and Jomphe [18]. Photographs of the above morphological features are presented in Irwin et al. [8].

### 3.4. Alborea Seed Parents 2n = 4x = 32, and near 32

The Alborea plants were selected from the cultigen Alborea [8] and were 2n = 32 based on pollen size and chromosome counts in an early generation [5]. The process involved 3 steps: 1. testing for hybrid production in backcrossing to *M*. *arborea*; 2. testing for 2n eggs; and 3. screening for 2n pollen (see Figure 1). Alborea 101 seed parent was selected from the Alborea 101 population that was developed by intercrossing the first hybrids of alfalfa X *M.arborea* reported in *Plants* 2013 [1]. The Alborea 301 series was developed by adding hybrids produced since 2013 to the 101 population by intercrossing. Alborea 101 was submitted to the Australian Pastures Genebank (APG) and catalogued as APG 84501 in 2018. The Alborea 301 series has a broader genetic base and was advanced one more generation before submitting it to the USDA Plant Introduction Program as PI 690775 in 2019.

The Alborea 101 plant and plants in the 301 series were selected for use because they were relatively self-sterile and had flowers that were cream or very light yellow. The light flower color enabled identification of hybrids with *M. arborea* by yellow flower color.

### 3.5. Octoploid Alfalfa Pollen Parents 2n = 8x = 64 Used in 2n Egg Test

Octoploid alfalfa plants were grown from remnant seed of earlier experiments [13,19]. The high chromosome number of 8x plants results in errors in both mitotic and meiotic cell divisions, requiring plants to be screened for acceptable pollen. About 50 plants were grown to flowering in the field to identify 8x plants with ample pollen that were large and relatively uniform. Five 8x plants had this type of pollen and were selected for use in 4x X 8x crosses undertaken in the greenhouse.

### 3.6. The Test for 2n Eggs in Alfalfa and Alborea Seed Parents

The test for 2n eggs is the same for alfalfa and Alborea, and alfalfa is used as the example. Pictures of female gametophytes in alfalfa with n and 2n eggs, respectively, have been published [13]. The test involved crossing 4x alfalfa X 8x alfalfa. The 8x pollen parent has n = 32 pollen and is described in *3.4* above. The 4x alfalfa plants have 2n = 4x = 32 chromosomes. Normal eggs have n = 16 chromosomes, and abnormal 2n eggs have 2n = 32. In the 4x X 8x cross, the union of normal eggs n = 16 X n = 32 normal pollen produces 48 chromosome 6x embryos, most of which abort due to endosperm failure [20]. Plants test positive for 2n eggs when the union of abnormal eggs 2n = 32 × n = 32 normal pollen produces a 64 chromosome 8x embryo, seed, and plant. Plants are grown to maturity and 8x plants are confirmed by pollen size.

The 4x X 8x crosses for 2n egg tests of both alfalfa and Alborea were undertaken in the winter of 2018–2019 in a greenhouse. Alfalfa progenies were grown in a field in 2019 and the Alborea were grown in a field in 2020. Plants were grown to maturity and 8x plants were identified by pollen size.

### 3.7. Test for 2n Pollen in Male Sterile and Self-Sterile Plants

The test for 2n pollen in male sterile plants appears counterintuitive and requires explanation. The male sterile alfalfa plants in this study have a few pollen grains in the anther, an example of this in 6–4 ms has been reported [21]. However, there is not enough pollen in male sterile plants to cause the anther to burst when the floret is tripped onto a bee or a pollen collection device. Hence, the plants are functionally male-sterile, and require external pollen to produce seed.

Pollen for microscopic examination is obtained by collecting anthers from approximately 10 florets by tripping the florets onto a weighing spatula, the tip of a pocketknife blade, or similar object. This produces a sticky clump of anthers and broken filaments which are scraped onto a slide in a drop of water or a stain, such as aniline blue in lactophenol. The clump in water or stain is macerated with a spatula to break the anthers and liberate pollen, a cover slip is applied, and it is then observed under a microscope at low power. The scant amount of pollen in the debris can be measured with an ocular micrometer. The 2n pollen grains are approximately one and a half times the diameter of n grains, as in Figure 1. Once 2 or 3 large pollen grains are observed and measured, the answer is yes, and counts are not necessary. Self-sterile plants, such as M8 or the Alborea parents, have a small amount of pollen that can be collected directly and observed as above. The relationship between pollen size and chromosome number was reviewed previously [9].

## 4. Conclusions

This is the third paper in a series in *Plants* on weakening and further weakening of the crossing barrier between alfalfa and *M. arborea*. The first level of weakening involved alfalfa parents that required large numbers of crosses to produce a hybrid. Further weakening was achieved by pedigree selection for alfalfa seed parents that produced hybrids more efficiently and by using Alborea seed parents. Most Alborea parents were efficient hybridizers. Seed parents in the first weakening produced 2n eggs and seed parents with further weakening had a higher frequency of 2n eggs. The frequency of 2n eggs was the same as the frequency of hybrids. Evidently, fertilization of 2n eggs was required to obtain hybrids by providing sufficient alfalfa genetic material for embryogenesis. Plants with a high frequency of 2n eggs produce more hybrids. Plants that produced 2n eggs also produced 2n pollen, leading to the conclusion that 2n pollen could be used to screen for seed parents for wide crosses. Finally, it was concluded that normal meiosis is the main barrier to interspecific hybridization in our material.

## Figures and Tables

**Figure 1 plants-12-00962-f001:**
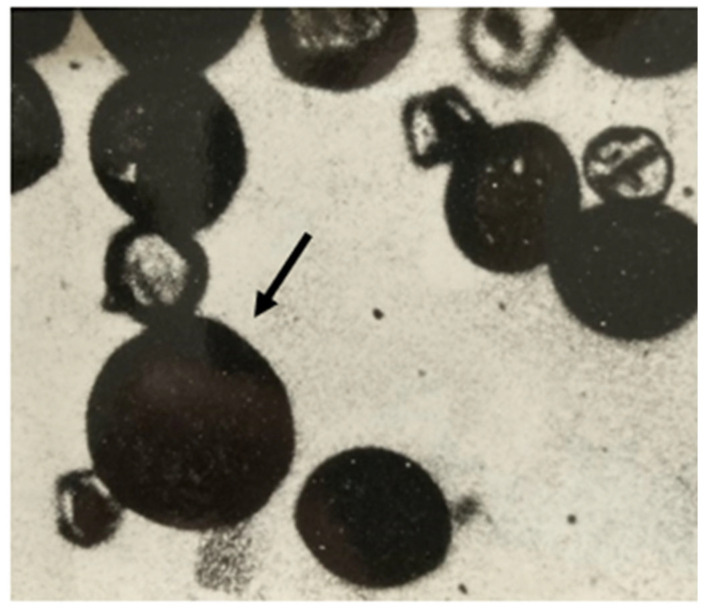
Large 2n pollen grain (arrow); remainder are n pollen grains of slightly variable size, and aborted grains from tetraploid Alborea 101 reported in Table 2 as producing 2n pollen. Alborea 101 is typical of all Alborea plants in this study. Photo X450, 1 cm = 40 microns.

**Table 1 plants-12-00962-t001:** Hybrids of alfalfa X *M. arborea*. Numbers of hybrids—numbers of florets pollinated in years 2003–2013 for alfalfa X *M. arborea*, and in years 2013–2022 for alfalfa X *M. arborea* and Alborea X *M. arborea* in Queensland and Wisconsin. See Section 3.3 for description of *M. arborea* pollen parents.

Year			
2003–2013		2013–2022	
Alfalfa Parents	Hybrids–Florets Pollinated–Florets per Hybrid	Alfalfa Parents	Hybrids–Florets Pollinated–Florets per Hybrid
**Queensland**		**Queensland**	
WA2071	5–2100–420	WA2570 (M8 S1)	13–500–38
WA2625 (Both MBms derived)	0–600	cv. Sequel (4 plants)	0–2000
**Wisconsin**		**Wisconsin**	
WI 6–4 ms	0–5000	WI 6–4 ms	0–500
WI (5 ms plants)	0–5000	WI M8	4–350–88
WI MBms	11–2700–250		
WI M8	16–1350–85	**Alborea Parents**	
		WA Alborea 2554	1–1000
		WA Alborea 2973 X	5–500–100
		Sequel	
		WI Alborea 101	3–40–13
		WI Alborea 301a	2–50–25
		WI Alborea 301b	1–50

**Table 2 plants-12-00962-t002:** Alfalfa and Alborea seed parents tested for 2n eggs in Wisconsin with pollen from 8x alfalfa (a 4x X 8x cross). Bold lettering indicates plants that produced hybrids with *M. arborea*.

Alfalfa Parents	Seed–Florets	Progeny 8x	Progeny 4x	Low Frequency of 2n Pollen *
6–4 ms	0–500	-	-	No
2 K–1 ms–1	6–100	2	3	Yes
2 K–1 ms–3	3–50	2	-	Yes
**MBms**	7–100	5	2	Yes
**M8**	12–100	9	2	Yes
	**Alborea Parents**			
**Alborea 101**	25–50	2	21	Yes
**Alborea 301a**	34–50	5	25	Yes
**Alborea 301b**	20–50	7	11	Yes
Alborea 301c	55–50	12	36	Yes
Alborea 301d	34–50	5	25	Yes

*: See Section 3.6 for how pollen is studied in male sterile (ms) plants.

**Table 3 plants-12-00962-t003:** *Medicago sativa* and subspecies *falcata* and *coerulea* used in the pedigree of M8. The *M. falcata* parents were from 4x cv. WISFAL [15,16] developed from 2x *M. falcata* plant introductions.

Subspecies	Description
*falcata*-1	Tall, robust, fall dormant, dark yellow flowers.
*sativa*-2	From Saudi-Arabian cultivar Wadi-Qura, winter active, purple flowers.
*falcata*-3	Like *falcata*-1, except thicker main stem that resists lodging.
*coerulea*-4	Purple flowered diploid, doubled with colchicine, used as a tetraploid.
*falcata*-5	Like *falcata*-1, except larger seed.
*sativa*-6	From USA cultivar Columbia 2000, fall dormant, purple flowered.
The crossing strategy was as follows: 2-way crosses 1 X 2 and 4 X 5. Followed by 3-way crosses (1 X 2) X 3 and (4 X 5) X 6. Finally, 3-way X 3-way cross involving the above.	

## Data Availability

All data are in paper.

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
