# Peer review of "The Hybridization Barrier between Herbaceous Medicago sativa and Woody M. arborea Is Weakened by Reproductive Abnormalities in M. sativa Seed Parents"

_plants, 2023, doi:10.3390/plants12040962_

Round 1

Reviewer 1 Report

The manuscript entitled "The Hybridization Barrier between Herbaceous Medicago sativa and Woody M. arborea is Weakened by Selection of Seed Parents II: Further Weakening" is interesting, well written and gives useful information regarding how the breeders can overcome the hybridization barrier between alfalfa and M. arborea.

However some improvements should be made.

As a general comment I dont agree with the characterization of this paper as a review paper since it gives and discusses the experimental data obtained by the authors.

Introduction only focuses on these two species (alfalfa and M arborea). I would suggest to give examples of other interspecies crossing where they observed 2n eggs.

References are rather too few.

Some tables need more clarifications (especially Table 2).

In the results and discussion section the writing is too condensed at some points and needs more explanations. 

Some minor points are corrected on the pdf file attached. 

Author Response

Thank you for your suggestions.  We accepted all suggestions except changing cultigen.  It is based on Wikipedia definition:  a plant that has been deliberately altered by humans.   We think it fits.  We are requesting change from review to article.  Added citations in other species where partial hybrids found, and 2n eggs suggested.  All your other markups are addressed.  Thanks again.

Reviewer 2 Report

This is an interesting report on methodology used to introgress M. arborea chromatin into M. sativa.  It would have been much easier to follow had the Methods and Materials section been presented before the Results and Discussion. There were several places where my questions were later answered in the M&M.  I realize it is technically difficult to document 2n eggs cytologically but it would have been a nice validation.  Minor changes in sentence structure are suggested.  The materials generated would be useful for further cytogenetics studies.  Perhaps other publications characterize the traits transferred from M. arborea to M. sativa and the usefulness of the germplasm.  I recommend this submission for publication with minor revision.  Comments and suggestions are in the attached pdf.

Author Response

We kept the MM after R & D, and added (see 3.1) etc. in R & D to help answer questions.  Added references in other species with results like ours.  Agreed with all your markups and made changes.  Backcrossing.  I was carried away with it.  Revised places where you had questions. Thank you.